# A Next-Generation Sequencing-Based Approach to Identify Genetic Determinants of Antibiotic Resistance in Cambodian *Helicobacter pylori* Clinical Isolates

**DOI:** 10.3390/jcm8060858

**Published:** 2019-06-15

**Authors:** Vo Phuoc Tuan, Dou Narith, Evariste Tshibangu-Kabamba, Ho Dang Quy Dung, Pham Thanh Viet, Sin Sokomoth, Tran Thanh Binh, Sok Sokhem, Tran Dinh Tri, Seng Ngov, Pham Huu Tung, Ngo Phuong Minh Thuan, Tran Cong Truc, Bui Hoang Phuc, Takashi Matsumoto, Kartika Afrida Fauzia, Junko Akada, Tran Thi Huyen Trang, Yoshio Yamaoka

**Affiliations:** 1Department of Environmental and Preventive Medicine, Oita University, Faculty of Medicine, Yufu 879-5593, Japan; vophuoctuandr@gmail.com (V.P.T.); evaristetshibangu@gmail.com (E.T.-K.); buihoangphuc412@gmail.com (B.H.P.); tmatsumoto9@oita-u.ac.jp (T.M.); kartikafauzia@gmail.com (K.A.F.); akadajk@oita-u.ac.jp (J.A.); 2Department of Endoscopy, Cho Ray Hospital, Ho Chi Minh 749000, Vietnam; quydung@gmail.com (H.D.Q.D.); bs.binh@yahoo.com.vn (T.T.B.); trantri73@gmail.com (T.D.T.); huutungbvcr@gmail.com (P.H.T.); minhthuan1177@yahoo.com (N.P.M.T.); tctruc@gmail.com (T.C.T.); 3Department of Endoscopy, Cho Ray Phnom Penh Hospital, Phnom Penh 12357, Cambodia; dr.narith@yahoo.com (D.N.); drsokhem@gmail.com (S.S.); 4Department of Integrated Planning, Cho Ray Hospital, Ho Chi Minh 749000, Vietnam; vietchoray@gmail.com (P.T.V.); sokomothsin@gmail.com (S.S.); 5Department of Cardiovascular Surgery, Cho Ray Phnom Penh Hospital, Phnom Penh 12357, Cambodia; 6Department of General Internal Medicine, Cho Ray Phnom Penh Hospital, Phnom Penh 12357, Cambodia; sengngovmd@gmail.com; 7Department of Molecular Biology, 108 Military Central Hospital, Hanoi 113601, Vietnam; huyentrang110@yahoo.com; 8Global Oita Medical Advanced Research Center for Health, Yufu 879-5593, Japan; 9Department of Medicine-Gastroenterology, Baylor College of Medicine, Houston, TX 77030, USA

**Keywords:** *Helicobacter pylori*, mutation, antibiotic resistance, next-generation sequencing, genetic determinants, whole genome sequence

## Abstract

We evaluated the primary resistance of *Helicobacter pylori* (*H. pylori*) to routinely used antibiotics in Cambodia, an unexplored topic in the country, and assessed next-generation sequencing’s (NGS) potential to discover genetic resistance determinants. Fifty-five *H. pylori* strains were successfully cultured and screened for antibiotic susceptibility using agar dilution. Genotypic analysis was performed using NGS data with a CLC genomic workbench. PlasmidSeeker was used to detect plasmids. The correlation between resistant genotypes and phenotypes was evaluated statistically. Resistances to metronidazole (MTZ), levofloxacin (LVX), clarithromycin (CLR), and amoxicillin (AMX) were 96.4%, 67.3%, 25.5%, and 9.1%, respectively. No resistance to tetracycline (TET) was observed. Multi-drug resistance affected 76.4% of strains. No plasmids were found, but genetic determinants of resistance to CLR, LVX, and AMX were 23S rRNA (A2146G and A2147G), GyrA (N87K and D91Y/N/G), and *pbp1* (P473L), respectively. No determinants were genetically linked to MTZ or TET resistance. There was high concordance between resistant genotypes and phenotypes for AMX, LVX, and CLR. We observed high antibiotic resistance rates of CLR, MTZ, and LVX, emphasizing the need for periodic evaluation and alternative therapies in Cambodia. NGS showed high capability for detecting genetic resistance determinants and potential for implementation in local treatment policies.

## 1. Introduction

Drug resistance is recognized as one of the greatest public health threats, leading to 700,000 deaths per year worldwide. This figure is expected to reach to 300 million people, with a loss of up to US$100 trillion for the global economy by 2050, if not addressed [1]. The issue is of great concern because of the rapid spread of multidrug-resistant strains, some of which may be untreatable with available treatments [2]. 

In this scenario, *Helicobacter pylori* (*H. pylori*) has emerged as an alarming bacterium due to its resistance rate, and it has been noted as one of 16 antibiotic-resistant pathogens that pose the most serious threat to human health, according to World Health Organization (WHO), since 2017 [3,4]. Along with being the most common bacteria that colonizes the human stomach, *H. pylori* is also known to cause a wide range of gastroduodenal diseases, including gastric adenocarcinoma. *H. pylori* elimination is thus one of the most effective strategies to reduce the *H. pylori*-related disease burden [5]. However, the emergence and rapid spread of drug-resistant *H. pylori* has become a major roadblock to eradicating the infection and a challenge for clinicians, because of a dramatic decline in the efficacy of classic first-line therapy, which has reached an unacceptable threshold (<80%) compared to other bacteria [6,7]. Therefore, many consensus conferences for *H. pylori* treatment (e.g., Toronto [8], Maastricht [9], Asia-Pacific [10], Kyoto [11]) have been held to attempt to deal with threats from the effectiveness of current eradication protocols caused by antibiotic resistance in different regions.

Cambodia is a developing country located in southeast Asia, a WHO region for which the standard triple therapy still remains effective because of a low clarithromycin (CLR) resistance rate [12]. Epidemiological studies have underlined the need of local information in different areas, because the distribution of resistance rates varies according to different populations and geographical regions [8,9,13]. While antibiotic resistance data has been reported in neighboring countries (e.g., Vietnam, Lao, Thailand, Myanmar, and Malaysia), information is still scarce in Cambodia [14]. Moreover, there is currently no Cambodian national guideline available for *H. pylori* treatment, and the choice of eradication therapy is mainly based on empirical decisions made by clinicians. Thus, an antibiotic susceptibility profile is needed in Cambodia to establish effective local guidelines for *H. pylori* eradication therapy and allocate appropriate public health policies against *H. pylori* antibiotic resistance-caused threats.

Antibiotic susceptibility testing is a reliable assay that provides useful information for clinicians to prescribe an appropriate *H. pylori* eradication regimen. However, this approach is not cost-effective, and is cumbersome and unfeasible in clinical practice. In fact, this method is not available for all clinical settings because of the special clinical microbiology laboratory requirements, like specific microaerobic culture conditions and a selective, nutrient-rich medium. Moreover, it is also time-consuming and laborious (about 10–14 days) because of *H. pylori*’s slow growth. As a result, it is hard to routinely perform *H. pylori* sensitivity testing in a clinic [15].

To deal with the drawbacks, a molecular method allowing identification of drug-resistant markers to promptly and accurately diagnose drug resistance is needed to improve *H. pylori* treatment, which is essential for risk stratification of patients and controlling drug resistance. The advent of molecular biology techniques (e.g., polymerase chain reaction (PCR), Sanger sequencing) has enabled the identification of molecular mechanisms underlying the observed phenotypic drug susceptibility testing (DST) to various antimicrobial agents. These conventional methods focused on specific mutations in a small region of the target gene. Some mechanisms of drug resistance are heterogeneous and some remain unclear and are still being explored, so these approaches limited the discovery of novel or rare resistance mechanisms [16]. Recently, the approach based on next-generation sequencing (NGS) has emerged as a comprehensive, cost-effective, and fast tool for disease surveillance, drug resistance prediction, and evolutionary analysis of infectious diseases [17,18]. For instance, NGS identified the species and drug susceptibility profile of *Mycobacterium tuberculosis* with an accuracy of up to 93%, and the cost of NGS screenings were 7% cheaper than routine clinical workflows (481£ vs. 518£, respectively). The results from NGS were also obtained faster than routine diagnosis (9 days vs. 21 days) [19]. However, there are specific challenges with *H. pylori* that have limited the use of these approaches for this bacterium. First, while other bacterial species have well-established comprehensive antibiotic resistance data like Antibiotic Resistance Gene-ANNOTation (ARG-ANNOT) [20], Comprehensive Antibiotic Resistance Database (CARD) [21], MEGAres [22], and ResFinder [23], these databases do not include much, if any, *H. pylori* data. Second, its enormous genetic diversity, due to high mutation and recombination rates, made it difficult to apply and interpret the results using approaches commonly employed in other bacteria to *H. pylori*. Here, we used a fast, computationally efficient, and accurate NGS-based approach employing a user-friendly graphical interface from the commercial software CLC genomic workbench that could circumvent these limitations. 

In this study, we aimed to characterize the primary resistance rate of five clinically used antibiotics including amoxicillin (AMX), CLR, metronidazole (MTZ), levofloxacin (LVX), and tetracycline (TET) from *H. pylori* clinical isolates in Cambodia. We also determined the corresponding genotypic molecular mechanism of each antibiotic using an NGS approach, and evaluated the correlation between genotypic and phenotypic DST. 

## 2. Materials and Methods

### 2.1. Study Design and Sampling Collection

This cross-sectional study included 206 consecutive outpatients (88 males and 188 females, mean age ± standard deviation 45.3 ± 15.3, age range 17–82 years) who had not been treated in relation to *H. pylori* previously. Patients had an esophagogastroduodenoscopic examination at the Endoscopy Department of Cho Ray Phnom Penh Hospital in Phnom Penh, Cambodia in March 2015. During the endoscopic procedure, gastric biopsy specimens were sampled from the antrum of each participant and used to isolate *H. pylori* using a standard previously-established culture [24]. Briefly, the gastric biopsy specimen was homogenized and inoculated on a *H. pylori*-selective plate (Nissui Pharmaceutical co., LTD, Tokyo, Japan) before being incubated for up to 10 days. The *H. pylori*-like colonies growing on the selective plates (clear, circular, purple, and convex) were identified using the biochemical reactions of catalase, oxidase, and urease, and by microscopic examination after Gram staining. They were sub-cultured for 3 to 4 days on Mueller Hinton II Agar plates (Becton Dickinson, Sparks, MD, USA) and supplemented with 10% horse blood (Nippon Biotest Laboratories Inc., Tokyo, Japan). All procedures were performed in microaerophilic conditions (10% O_2_, 5% CO_2,_ and 85% N_2_) and incubated at 37 °C. A total of 55 *H. pylori* clinical strains from 55 separate patients (23 males and 32 females, age ± standard deviation 43.9 ± 13.7, age range 19–76 years) were successfully isolated from the gastric biopsy specimens. 

### 2.2. Ethical Considerations

The study protocol was approved by the ethical committees of Oita University, Japan and Cho Ray Phnom Penh Hospital, Cambodia. All patients recruited signed informed consent.

### 2.3. Phenotypic Antibiotic Susceptibility Testing

The antibiotic susceptibility phenotype was defined using the agar dilution assay to determine the minimum inhibitory concentrations (MICs) of antibiotics following the protocols of the Clinical and Laboratory Standards Institute (Wayne, PA, USA) [25]. A set of five common antibiotics were used in the experiment: AMX, MNZ, CAM, TET, and LVX (Wako Pure Chemical Industry, Osaka, Japan). Briefly, bacteria were sub-cultured for 3 days on Mueller Hinton II Agar plates supplemented with 5% horse blood and a two-fold serial dilution of antibiotics. The bacterial suspension, adjusted to a McFarland opacity standard of 0.5, was inoculated onto the plates. The MICs of antibiotics were determined after 72 h of incubation. *H. pylori* strain 26695 was used as the control strain. Clinical breakpoints were used for discriminating between susceptible and resistant phenotypes following the guidelines of the European Committee on Antimicrobial Susceptibility Testing (EUCAST; available in http://www.eucast.org/). The resistance to antibiotics was defined as MICs exceeding 0.125 mg/L for AMX, 0.25 mg/L for CLR, 8 mg/L for MTZ, and 1 mg/L for TET and LVX. The multidrug-resistant (MDR) phenotype was defined as the phenotype that was resistant to two or more antibiotic classes while single-drug resistance (SDR) was resistance to only one of five antibiotics used in the assay. 

### 2.4. Genotype Analysis of Antibiotic Susceptibility

#### 2.4.1. DNA Extraction, Library Preparation, and Whole Genome Sequencing 

The total genomic DNA of isolates was extracted using the QIAamp DNA Mini Kit (QIAGEN, Crawley, UK), quantified using the Quantus Fluorometer (Promega, Madison, WI, USA), and sequenced using the Miseq platform (Illumina, Inc., San Diego, CA, USA). DNA libraries were prepared using the Nextera XT DNA sample kit (Illumina, San Diego, CA, USA), allowing paired-end sequencing of 300-bp reads.

#### 2.4.2. Bioinformatic Analysis to Identify Genetic Determinants of Antibiotic Resistance

##### Analysis of Drug Susceptibility Genotypes Encoded by Plasmid in *H. pylori* Isolates

Raw short read data from *H. pylori* clinical isolates were used to investigate the known plasmids encoding antibiotic resistance using PlasmidSeeker with 26695 (NC_000915.1) as the reference genome [26]. The read coverage of short DNA oligomers with length k (k-mer) against an available plasmid reference database (http://bioinfo.ut.ee/plasmidseeker/) was analyzed, and the k-mer abundance was used to distinguish between plasmid and bacterial sequences. A plasmid was considered present with at least 80% of k-mers found.

##### Analysis of Drug Susceptibility Genotypes Encoded by Genomic DNA of *H. pylori* Isolates

To reduce the false discovery rate of single nucleotide polymorphisms (SNPs), the low-quality bases (the quality of bases <Q30) and adapters were trimmed using Trimmomatic v0.30 [27]. The filtered high-quality reads were assembled to produce contigs using SPAdes v3.13.0 [28]. The draft genome assembly quality was assessed using QUAST v5.0.2 [29]. All genomes sequenced in this study were deposited at National Center for Biotechnology Information (NCBI) under BioProject ID PRJNA547954.

To assess the genetic relatedness between *H. pylori* strains, a core SNP alignment from draft genome mapping against reference strain 26695 was generated by using Snippy v.3.2 [30]. Recombinant regions were predicted and removed using Gubbinsv2.3.4 [31]. A maximum likelihood tree was constructed by RAxML v8.2.9 with a general-time reversible nucleotide substitution model, applying a GAMMA correction for site variation [32]. 

The antibiotic susceptibility genotypes were assessed in *H. pylori* resistance-related genes using the CLC genomic Workbench v8.5.1 (CLC Bio, Aarhus, Denmark). Using the map read to reference function, the trimmed reads were first directly mapped against a custom database of target gene sequences that likely play a biological role in antibiotic resistance in *H. pylori* species. These targeted genes were retrieved from the 26695 reference genome, including *gyrA* (*HP0701*) for LVX, *rdxA* (*HP0954*) for MTZ, *pbp1* (*HP0597*) for AMX, 23S rRNA for CLR, and 16S rRNA for TET resistance. Variant identification between each isolate sequence and reference gene was performed using the CLC basic variant detection function with default options. These variants were summarized and assessed with statistical approaches for their eventual association to the resistance phenotype using the CLC Fisher exact test function. The nomenclature describing genetic variants was followed as previously reported [33]. For instance, N87K denotes the N (asparagine) to K (lysine) substitution at codon position 87. The presence or absence of the genetic determinants of antibiotic resistance, the phenotypic resistance patterns, and the phylogenetic tree were illustrated as heat maps using Phandango [34].

### 2.5. Statistical Analysis

All statistics and data visualization were performed in R Environment version 3.5.1. The proportions of phenotypic DST and its 95% confidence intervals (CIs) were calculated using the Clopper-Pearson exact method from ‘DescTools’ package [35]. The association of mutations or SDR/MDR with MIC levels was analyzed using the Mann-Whitney *U* test. Genotype-phenotype agreements were analyzed using Cohen’s kappa statistics. A kappa coefficient (*k*) value of < 0.4, 0.4–0.6, 0.61–0.8, and 0.81–1.0 indicated low, moderate, substantial, and perfect agreement, respectively. A significance of *p* < 0.05 using a two-tailed test was established as statistically significant. 

## 3. Results

### 3.1. H. pylori Primary Antibiotic Susceptibility in Cambodia

The antibiogram of 55 isolates from Cambodia is shown in Table 1. The range and distribution of MIC values for each antibiotic are shown in Figure 1. We noted a resistance profile with the highest prevalence of resistance to MTZ 96.4% (95% CI, 87.5–99.6), followed by LVX 67.3% (95% CI, 53.3–79.3), CLR 25.5% (95% CI, 14.7–39), and finally, AMX 9.1% (95% CI, 9.1–30.9). No TET-resistant isolates were identified. Only two strains (2/55, 3.6%) were susceptible to all antibiotics tested.

Notably, 21.8% (95% CI, 11.8–35) were SDR, and all of them had MTZ resistance. The rate of MDR was high, up to 76.4% (95% CI, 63–86.8). Among all clinical isolates, dual drug resistance to LVX + MTZ and MTZ + CLR was found in 40% (95% CI, 27–54.1) and 7.3% (95% CI, 2–17.6) of isolates, respectively. Triple resistance to AMX + LVX + MTZ and CLR + LVX + MTZ accounted for 9.1% (95% CI, 3–20) and 18.2% (95% CI, 9.1–30.9), respectively. The average MDR MIC was significantly higher than SDR MIC in LVX (8.6 mg/L vs. 0.31 mg/L, *p* = 0.01) and CLR (3.15 mg/L vs. 0.04 mg/L, *p* = 0.003). In addition, MTZ (81.2 mg/L vs. 69.3 mg/L) and AMX (0.18 mg/L vs. 0.06 mg/L) also showed higher mean MICs in MDR compared to SDR.

### 3.2. Relationship between Antibiotic Resistance Patterns and Strain-Relatedness in Cambodian H. pylori Isolates

The draft genomes of 55 *H. pylori* isolates were assembled, and the assembly qualities are shown in Appendix A. Two isolates (KH0155 and KH0195) with unusual genome sizes were excluded. Hence, 53 *H. pylori* genomes could be used in further analysis. When reconstructing a maximum likelihood tree of core SNPs with respect to different resistance patterns in Figure 2, no strain-relatedness was found to be associated to either each single antibiotic or SDR/MDR. Alternatively, these strains had distinct susceptibility profiles regardless of genetic relatedness between strains.

### 3.3. Plasmid Detection from Cambodian H. pylori Isolates

A plasmid reference database including 8514 known plasmids from published species, 41 of which are specific for *H. pylori,* were used to determine their presence. No probable plasmids encoding a resistance gene were found among 53 analyzed *H. pylori* genomes (Appendix A). 

### 3.4. Genetic Determinants of Cambodian H. pylori Antibiotic Resistance

#### 3.4.1. CLR Resistance

We investigated genetic variations in the full-length 23S rRNA gene, in which the single point mutation in the peptidyl transferase region of domain V of the gene is known to be responsible for CLR resistance [36]. Because of the presence of two copies of the 23S rRNA gene in the *H. pylori* genome, we examined the depth of reads mapped to each nucleotide variation position that could detect allelic variation between the two copies. Among the 13 CLR-resistant and 40-susceptible strains, we identified 67 variants. Only two mutations at nucleotide positions A2146G (*p* = 0.01) and A2147G (*p* = 0.000001) were significantly associated with CLR resistance (Appendix A). CLR-resistant strains had 23.1% (3/13) G-2146 and 61.5% G-2147, respectively. All CLR-susceptible strains (40/40, 100%) possessed A-2146 and A-2147 alleles. In addition, strains harboring mutations at either A2146G or A2147G showed significantly higher MICs than those without mutations (all *p* < 0.05). The average MICs were not significantly different between strains with mutations at position 2146 or 2147 (Figure 3A).

While examining the depth of reads mapped, 9 CLR-resistant strains had all reads (100%) mapped to either nucleotide position G-2146 or G-2147, indicating that the mutations occurred in both copies of 23S rRNA. Only two CLR-resistant strains had heterogeneous alleles between two copies of 23S rRNA, indicating that the mutations occurred in one of two copies of 23S rRNA, namely KH0097 (29.6% reads mapped to G at nucleotide position 2146) and KH0113 (13.9% reads mapped to G at nucleotide position 2147). 

#### 3.4.2. LVX Resistance

We assessed GyrA, which is known to confer *H. pylori* LVX resistance via point mutations in the quinolones resistance-determining region (QRDR) [36]. The genetic variations within the full-length GyrA of 36 LVX-resistant and 17-susceptible strains were explored. Among the 128 variants identified, only mutations at codon positions N87K (*p* = 0.005) and D91Y/N/G (*p* = 0.00004) were significantly associated with LVX resistance (Appendix A). Interestingly, 33.3% (12/36), 13.9% (5/36), 13.9% (5/36), and 27.8% (10/36) of LVX-resistant strains had Lys-87, Tyr-91, Gly-91, and Asn-91, respectively. All LVX-susceptible strains (17/17, 100%) possessed Asn-87 and Asp-91. 

Among 36 LVX-resistant strains, 30 (83.3%) strains had mutations at codon 87 or 91. On the other hand, six strains did not show any mutations at these codons, but they were resistant. Overall, strains harboring mutations at either codon 87 or 91 showed significantly higher MICs than those without mutations (all *p* < 0.05). Interestingly, strains with double mutations at codon 87 and 91 showed significantly higher MICs than those with single mutations at either codon 87 or 91 (*p* < 0.05) (Figure 3B).

#### 3.4.3. AMX Resistance

We examined the genetic variants of 5 AMX-resistant and 48-susceptible strains within the full-length *pbp1*. Mutations in the gene are thought to be the major factor for AMX resistance [37,38]. We identified 45 variants. Three codons E406K (*p* = 0.005), P473L (*p* = 0.0004), and T593A/G/K (*p* = 0.005), were significantly associated with AMX resistance (Appendix A). P473L was found in 3/5 (60%) of AMX-resistant strains, but it was not observed in AMX-susceptible strains. Although E406K and T593A/G/K were predominant in AMX-resistant strains, they were also detected in susceptible strains (Figure 2). 

#### 3.4.4. MTZ Resistance 

We investigated *rdxA*, in which the loss of function (e.g., nonsense or frameshift mutations) in the gene result in protein mistranslation or truncation, which is known to play a crucial role in MTZ resistance [39]. We found that 30 out of 51 (58.8%) MTZ-resistant strains had an *rdxA* truncation, including 15 (50%) frameshift and 15 (50%) nonsense mutations. Both MTZ-susceptible strains had an intact *rdxA* encoding a full-length functional *rdxA* (Appendix A, Figure 2). However, statistical significance was not detected by comparing the genetic variants between MTZ-resistant (*n* = 51) and MTZ-susceptible strains (*n* = 2).

#### 3.4.5. TET Resistance

We examined the 16S rRNA gene to discover mutations reported to confer TET resistance, specifically A926G and A928C [40]. No strains showed any nucleotide substitutions at position A-926 or A-928C, agreeing with phenotypic susceptibility observed in all strains (Figure 2). 

### 3.5. Comparison between Antibiotic Susceptibility Genotypes and Phenotypes 

To determine the utility of whole genome sequencing (WGS) in predicting phenotypic drug susceptibility to different antibiotics, we compared the antibiotic resistance phenotypes with their corresponding genotypes. As shown in Table 2, the concordance between genotypic and phenotypic drug resistance was in perfect agreement for CLR (kappa = 0.89) and substantial agreement for LVX (kappa = 0.73) and AMX (kappa = 0.73).

## 4. Discussion

Despite an increasing worldwide trend in antibiotic resistance, a recent meta-analysis showed that only southeast Asia (10%) and the Americas (10%) appeared to have low CLR resistance rates compared to all WHO regions [12]. In addition, southeast Asia also displayed the lowest CLR resistance rate in the Asia-Pacific region [14]. These data suggested that CLR-based triple therapy might still be useful as a first-line treatment in this area. However, the accumulated data also found that there was increasing resistance to CLR, MTZ, and LVX over different time periods in the Asia-Pacific region: before 2000, 2000–2005, 2006–2010, and 2011–2015 [14]. Indeed, a recent report from Singapore showed a changing CLR resistance profile over 15 years (7.9–17.1%), suggesting that there was increasing CLR resistance in southeast Asia [41]. Consistent with the general context, our study revealed a relatively high prevalence of CLR resistance (25.5%) in Cambodia. This is an alarming sign of the increasing CLR-resistant rate in South-East Asia.

A survey collected expert opinions about *H. pylori* in the Association of Southeast Asian Nations (ASEAN) and declared that the resistant patterns in ASEAN countries were characterized by high CLR and MTZ resistance, as observed in this study with 40% strains being CLR + MTZ dual resistant [42]. In agreement with neighboring countries like Vietnam (70%) and Thailand (70%), our study also showed extremely high MTZ resistance (96.4%) in Cambodia. The high MTZ resistance might be caused by common prescriptions for parasitic infections (particularly in tropical countries), pelvic inflammation, or dental infections [14]. De Francesco V et al. found that high MIC values in MTZ-resistant (MIC > 32 mg/L) and CLR-resistant strains (MIC > 8 mg/L) were relevant to an in vivo resistance pattern. Therefore, they are a potential marker for predicting the cure rate [43]. Likewise, even though CLR resistance in our study was 25.5%, about half of this was found to have high MIC resistance to both MTZ and CLR, suggesting that the administration of standard triple therapy as a first-line treatment might no longer be effective, particularly in Cambodia, and it should be applied with caution based on local susceptibility data.

LVX has recently been prescribed as a salvage therapy in patients with first-line therapy failure [9]. However, we observed a high proportion of LVX resistance (67.3%), even in the primary resistance rate in Cambodia. This implicated the higher secondary resistance rate to LVX, and this may reduce the efficacy of the LVX-based regimen. The high LVX resistance might be caused by frequent use of LVX to treat urinary and respiratory infections [14]. The fact that all strains were susceptible to TET is vital for guiding second-line regimens with potential TET-based or quadruple therapies in Cambodia. Additionally, the number of multidrug-resistant strains in this study appeared to be a serious challenge (76.4%) in the fight against infections and a hindrance to the success of eradication regimens. This has been attributed to widespread antibiotic misuse in the country because of habitual antibiotic-seeking behaviors, uncontrolled use of antibiotics, or the ease of accessing antibiotics in Cambodia [44,45]. A national policy to reduce the use and abuse of antimicrobials should be established in Cambodia.

Bacteria use two main genetic strategies to respond to antibiotic attack. One is the result of mutations in genes associated with the mechanism of action of the drug, and the second is due to the acquisition of external genetic resistance elements via horizontal gene transfer [46]. In the latter, a plasmid is one of the most important factors in disseminating resistance genes between bacteria. Our results showed that there was no plasmid found in all strains, suggesting that drug resistance in *H. pylori* is mainly based on the former. Unlike the other bacteria, an approach based on point mutations in the genomes to identify genetic determinants of *H. pylori* resistance is useful and less complex [47].

Because of greater speed and accuracy, molecular susceptibility testing has become a promising alternative choice to replace phenotypic DST [16]. The most common approach is based on Sanger sequencing, in which the mutations conferring resistance are determined by comparing the sequence of a small region of a target gene in resistant strains with a given reference strain or a small group of sensitive strains [48]. This approach has revealed novel variants compared to a reference strain. However, these variants might be caused by differences in *H. pylori* population structures or mutational effects on fitness (so-called compensatory mutations) that do not confer resistance [49]. Therefore, we used an approach focusing on the strength of associations between all possible genetic variants within the target gene and trait of interest. Intriguingly, our approach preserved the well-known mutations in GyrA (N87K or D91Y/N/G) for LVX resistance and mutations in 23S rRNA (A2146G or A21427) for CLR resistance. A previous study showed that LVX and CLR appeared to have high concordance between genotypic and phenotypic DST, so the mutations in these genes (GyrA for LVX and 23S rRNA for CLR resistance) were used to develop molecular methods to rapidly detect genetic determinants like GenoType HelicoDR [50]. This tool allows simultaneous detection of 23S rRNA and GyrA mutations within 6 h. It has been suggested that a molecular method to detect LVX and CLR resistance could be applied in Cambodia, to assist clinicians making therapeutic decisions. Further studies to validate these tools are needed.

Additionally, our approach also considered the number of mutated gene copies like in the case of the 23S rRNA. *H. pylori* has two copies of the gene [51]. It is still unknown whether the presence of a mutation on either a single copy or both copies is needed to cause CLR resistance. In our study, we found two CLR-resistant strains, one with A2146G and the other with A2147G in only one copy of 23S rRNA. There were no sensitive strains harboring the heterogeneous alleles. Similar to the case of *Streptomyces ambofaciens*, where one of four rRNA alleles caused macrolide resistance, it was suggested that changes in one of the two alleles could be sufficient to confer *H. pylori* CLR resistance [52].

The mechanism of AMX resistance is unclear, but it was believed to be caused by a point mutation in the *pbp1A* gene [38]. Some genetic determinants (e.g., N562Y, S414R, T593A, G595S, and A599T) have been reported using a classic approach, in which the limitation was demonstrated as discussed above. Because of the invalidity of these mutations in other studies, a recent paper by Chihiro K et al. [53] concluded that AMX resistance determination should not be identified based on one amino acid substitution at a specific position. Our study found three AMX-resistant associated loci (P473L, E406K, and T593A/G/K mutations). All the changes were located adjacent to the motifs confirmed to be responsible for AMX resistance (SAIK_368–371_, SLN_433–435_, SNN_559–561_, KTG_555–557_) using experimental study [54]. P473L is a potential mutation associated with AMX resistance. Indeed, this amino acid substitution was found in 60% of AMX-resistant strains, but no AMX-sensitive strains possessed the mutation. Future study using natural transformations is needed to validate this result. The 40% of unexplained resistant strains might be interesting to target to identify new mechanisms of AMX resistance. On the other hand, the mutations E406K and T593A/G/K might play a minor role in AMX resistance, or just be compensatory mutations alleviating the fitness costs of a resistance mutation, based on their presence in both sensitive and resistant strains [49]. Interestingly, Kwon YH et al. compared 77 AMX-resistant strains and 77 AMX-sensitive strains and found D479E and T593 mutations associated with AMX resistance [37]. Even though the latter was confirmed by natural transformation, it was also suggested that this change alone might not be enough to express resistance, because the MICs of AMX-resistant transformants were lower than naturally-occurring AMR-resistant isolates.

Resistance to MTZ primarily involves the inactivation of the *rdxA* gene encoding an oxygen-insensitive nicotinamide adenine dinucleotide phosphate hydrogen (NADPH) nitroreductase that catalyzes the reduction of MTZ [39]. Our study revealed that the truncated *rdxA* is responsible for 58.8% of MTZ-resistant isolates in Cambodia. Chua EG et al. recently reported that a R16H/C substitution of *rdxA* is significantly associated with MTZ-resistant strains [55]. However, we found that 100% (2/2) of MTZ-susceptible strains and 29.4% (15/51) of MTZ-resistant strains possessed this mutation. Because of the small number of MTZ-susceptible strains included in the study, further study is necessary to validate the role of R16H/C substitution conferring MTZ resistance. Similarly, all strains in this study were susceptible to TET. Interestingly, no strains had mutations at positions A926G or A928C in the 16S rRNA.

This is the first study to report *H. pylori* antibiotic-resistant patterns in Cambodia to provide an overview of antibiotic resistance conditions in the country. This could provide direction for national guidelines or protocols for *H. pylori* eradication. However, our study also had several limitations. First, this is a hospital-based study, so the result does not reflect the resistance patterns in the general population. Nevertheless, our study also revealed reliable, accurate resistance markers to predict phenotypic DST, especially in AMX, CLR, and LVX. This could be applied as rapid alternative investigation without culture in a clinic. Second, the sample size is relatively small for some antibiotics. Therefore, there were only few (AMX) or no resistant strains (TET) that might affect the strength of association. In addition, we only determined the genetic determinants for antibiotic resistance within well-known genes in this study. However, we provided high-quality genomes with clear clinical information, so future studies at genome level, like bacterial genome-wide association studies (GWAS), can provide further insights into the mechanism of drug resistance together with the other *H. pylori* genomes [56]. Finally, our study revealed the resistance rates of commonly used antibiotics in vitro. Clinical trials are needed to confirm the efficacy of these relevant regimens.

## 5. Conclusions

In conclusion, *H. pylori* primary resistance rates to CLR, MTZ, and LVX were high in Cambodia, warning that the standard triple therapy as a first-line treatment may no longer be effective in southeast Asia, particularly in Cambodia. A periodic evaluation is essential to closely monitor and control the antibiotic resistance status in this area. Our results suggest reliable genetic markers that can be used as a rapid and accurate detection test for *H. pylori* resistance, especially for CLR, LVX, and AMX resistance. Additionally, our data suggested that a next-generation sequencing based approach is a useful, reliable, and reproducible method to detect antibiotic resistance-related genetic determinants. We also noted that future studies using a bacterial GWAS approach with careful preparation of the dataset might provide further insights about the mechanism of drug resistance in *H. pylori*.

## Figures and Tables

**Figure 1 jcm-08-00858-f001:**
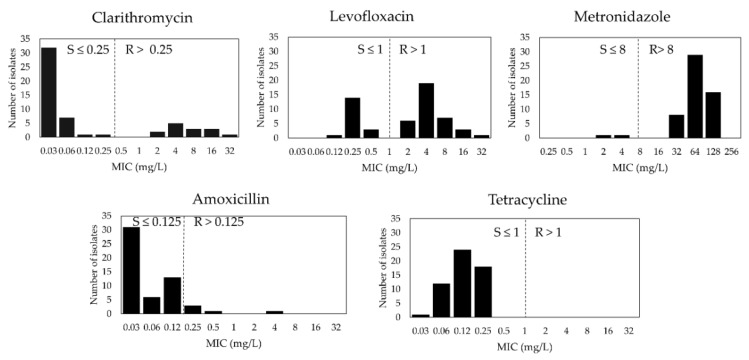
Antibiotic susceptibility testing based on agar dilution test from 55 isolates collected in Cho Ray Phnom Penh Hospital, Phnom Penh, Cambodia. Clinical breakpoints according to European Committee for Antimicrobial Susceptibility Testing (EUCAST) were drawn as black dashed line. R denotes resistance and S denotes susceptibility, respectively.

**Figure 2 jcm-08-00858-f002:**
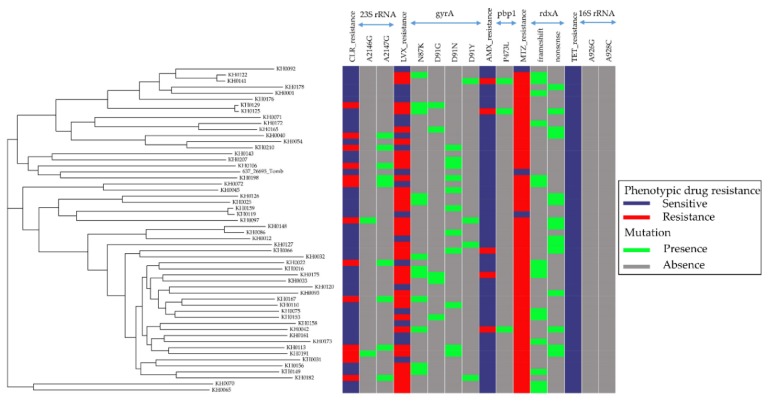
Genomic analysis of 53 *H. pylori* genomes sequenced in this study and the 26695 reference strain. Maximum likelihood phylogenetic tree based on whole genome sequences, the resistance patterns, and their corresponding genetic determinants for each antibiotic. The sensitive and resistant patterns are denoted by dark blue and red rectangles, respectively. The presence and absence of mutations are denoted by green and grey rectangles, respectively.

**Figure 3 jcm-08-00858-f003:**
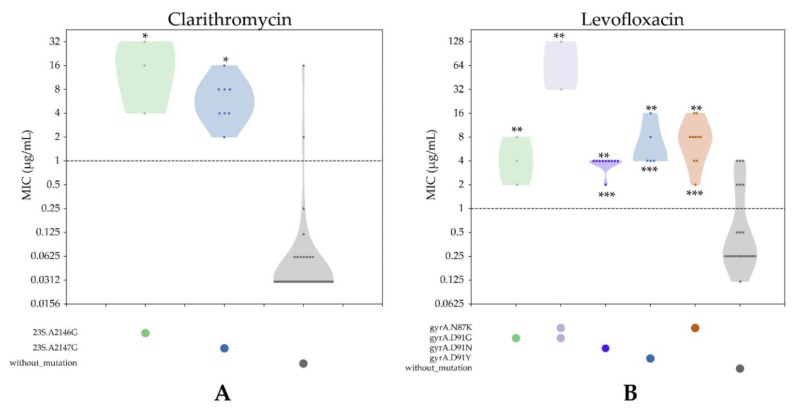
Distribution of minimum inhibitory concentrations (MICs) and corresponding genetic determinants of resistance. Dotted horizontal lines mark clinical breakpoints. (**A**) Distribution of MICs for clarithromycin (CLR) of all observed combinations of relevant antibiotic-resistant determinants. * indicates *p* < 0.05 compared to groups without mutation. (**B**) Distribution of MICs for levofloxacin (LVX)s for all observed combinations of relevant antibiotic-resistant determinants. ** indicates *p* < 0.05 compared to groups without mutation. *** indicates *p* < 0.05 compared to groups with double mutation at codons 87 and 91.

**Table 1 jcm-08-00858-t001:** Resistance pattern of 55 *H. pylori* strains in Cambodia.

Resistance Pattern	Number of Strains	Percentage % (95%CI)
Susceptible to all	2	3.6 (0.4–12.5)
All resistance	53	96.4 (87.5–99.6)
CLR	14	25.5 (14.7–39)
LVX	37	67.3 (53.3–79.3)
AMX	5	9.1 (3–20)
MTZ	53	96.4 (87.5–99.6)
TET	0	0 (0–6.5)
Mono Resistance	12	21.8 (11.8–35)
CLR only	0	0 (0–6.5)
LVX only	0	0 (0–6.5)
AMX only	0	0 (0–6.5)
MTZ only	12	21.8 (11.8–35)
TET only	0	0 (0–6.5)
Multiple resistance	42	76.4 (62.9–86.8)
LVX + MTZ	22	40 (27–54.1)
CLR + MTZ	4	7.3 (2–17.6)
AMX + LVX + MTZ	5	9.1 (3–20)
CLR + LVX + MTZ	10	18.2 (9.1–30.9)

Abbreviations: clarithromycin (CLR), levofloxacin (LVX), amoxicillin (AMX), metronidazole (MTZ), tetracycline (TC).

**Table 2 jcm-08-00858-t002:** Comparison between genotypes and phenotypes of antibiotic resistance.

Antibiotic	Susceptible Phenotype		Resistant Phenotype		Agreement	Kappa Coefficiency	*p*-Value
	Resistant Genotype	Susceptible Genotype	Resistant Genotype	Susceptible Genotype			
AMX	0	48	3	2	96.20%	0.73 (95% CI, 0.47–0.99)	3.29 × 10^−8^
MTZ	0	2	30	21	60.40%	0.1 (95% CI, 0.02–0.21)	0.1
LVX	0	17	29	7	86.80%	0.73 (95% CI, 0.47–0.99)	3.81 × 10^−8^
CLR	0	40	11	2	96.20%	0.89 (95% CI, 0.62–1.00)	6.35 × 10^−11^
TET	0	53	0	0	-	-	-

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
