# Peer review of "A Next-Generation Sequencing-Based Approach to Identify Genetic Determinants of Antibiotic Resistance in Cambodian Helicobacter pylori Clinical Isolates"

_jcm, 2019, doi:10.3390/jcm8060858_

Reviewer 1 Report

In this manuscript the authors describe the antibiotic resistance profile of 55 H. pylori strains isolated from patients where anti microbial treatment failed to eradicate H. pylori. Next generation sequencing was used to genotype the strains and identify known mutations associated with antibiotic resistance. These mutations were correlated with phenotypic analysis. Very high rates of resistance to metronidazole and levofloxacine were found and moderate levels of resistance to  clarithromycin and and amoxicillin were observed. There was a good correlation between resistant genotypes and phenotypes for amoxicillin, clarithromycin and levofloxacine. While the correlation between genetic marker for resistance to metronidazole and phenotype did not reach statistical significance this may be due to the low numbers of strains tested that were sensitive. No resistance to tetracycline was observed and this correlated with no genetic markers for tetracycline observed.

This information is useful for prescribing in Cambodia and in developing local guidelines around prescribing and prevention of anti-microbial resistance. It also suggests that next generation sequencing may be a useful methodology for prediction of anti-microbial resistance or sensitivity.

While the study appears to have been carefully done there are some details that need to be included in the methodology namely the details of patients that the strains were isolated from.

All patients had failed treatment for H. pylori infection. How was this tested?

What was the initial treatment prescribed for H. pylori infection in this cohort?

We are not told how many patients were included in the study and how many H. pylori were isolated from.

Were the 55 strains included in this study from 55 separate patients or were multiple strains isolated from some patients?

If multiple strains were isolated from some patients did the antibiotic resistance profile of these strains differ from one another? This would have important implications for treatment strategies.

Did sequencing confirm that each of the 55 isolates were seperate strains or were there some strains shared between patients?

The introduction needs to include a description of what the standard treatment strategy used for H. pylori infection is in Cambodia.

The discussion needs to include what the general prescribing patterns for mentronidazole and levofloxacine are in Cambodia and what other organisms are resistant to these drugs.

There is also a need to compare the cost of sequencing with traditional phenotypic testing for anti-microbial sensitivity.

The English in the manuscript needs to be improved. In places it is difficult to understand the sense of the sentences. The whole manuscript needs to be checked with a native English speaker.

Author Response

Response to comments raised by Reviewer 1

Comment 1. While the study appears to have been carefully done there are some details that need to be included in the methodology namely the details of patients that the strains were isolated from.

Response: We thank you for your comment. We added the information about patient characteristics for the collected samples. Please kindly find this information in the Materials and Methods section in page 3, lines 115-116 and 128.

Comment 2. All patients had failed treatment for H. pylori infection. How was this tested?

Comment 3. What was the initial treatment prescribed for H. pylori infection in this cohort?

Response: We thank you for these two questions. In this study, we aimed to describe the primary resistance in patients who had not been eradicated of H. pylori before inclusion in the study. In addition, we did not follow-up with patients for the success of any eventual H. pylori treatment. In the absence of guidelines for H. pylori eradication, the treatment options in Cambodia rely on the personal experiences of each clinician. Because of this, it is thus to first focus on the status of primary antibiotic resistance patterns of H. pylori isolates circulating in Cambodia. This study yielded initial information useful to contribute to evaluate and draw up the first-line treatment therapy in Cambodia. At the same time, the idea of evaluating the actual eradication success in cohorts of Cambodian patients is valid, and would come in our future projects.

Comment 4. We are not told how many patients were included in the study and how many H. pylori were isolated from.

Response: We thank you for this comment. We added the information in the Materials and Methods section in page 3, lines 115-116 and 128.

Comment 5. Were the 55 strains included in this study from 55 separate patients or were multiple strains isolated from some patients?

Comment 6. If multiple strains were isolated from some patients did the antibiotic resistance profile of these strains differ from one another? This would have important implications for treatment strategies.

Response: We thank you for these two questions. 55 strains included in this study were isolated from 55 patients. We clarified this in page 3, line 128.

Comment 7. Did sequencing confirm that each of the 55 isolates were seperate strains or were there some strains shared between patients?

Response: We thank you for your question. Based on the phylogenetic tree analysis as shown in Figure 2, the 55 strains were not related each other and were not shared between patients.

Comment 8. The introduction needs to include a description of what the standard treatment strategy used for H. pylori infection is in Cambodia.

Response: We thank you for your comment. As mentioned in Introduction section page 2, lines 73-74, there was no standard treatment strategy applied by Cambodian clinicians because no Cambodian official guidelines for H. pylori eradication were established. The therapeutic decisions made by clinicians for H. pylori eradication relied mostly on their personal experiences.

Comment 9. The discussion needs to include what the general prescribing patterns for mentronidazole and levofloxacine are in Cambodia and what other organisms are resistant to these drugs.

Response: We thank you for your comment. However, there were no published data about the usage of metronidazole and levofloxacin in Cambodia. There were only two reports warning about the widely uncontrolled use and misuse of antibiotics in Cambodia that are cited on references 44-45 on page 9, line 338 of the manuscript. To explain why the resistance rate is high for metronidazole and levofloxacin, we added more information on page 9, lines 320-321 and 331-332.

Comment 10. There is also a need to compare the cost of sequencing with traditional phenotypic testing for anti-microbial sensitivity.

Response: We thank you for this comment, and it raises an interesting question. Unfortunately, it was out of the scope of our current study. This issue can be the focus of our future plans. However, this issue was reported in other bacteria like Mycobacterium tuberculosis instead of H. pylori. We added a comment regarding this issue on page 3, lines 96-99.

Comment 11. The English in the manuscript needs to be improved. In places it is difficult to understand the sense of the sentences. The whole manuscript needs to be checked with a native English speaker.

Response: Thank you for your advice. Our manuscript was checked by a professional English editing service (Honyaku Center Inc. http://www.honyakucenter.jp/).

Reviewer 2 Report

               The study of Tuan et al. focuses on studying the drug-resistance Helicobacter pylori strains within the Cambodia population in order to identify if the current therapy is effective in eradication pathogen.

The research group isolated 55 strains from the patients, but in the material and method section, however, there is absent of information from how many patients samples were obtained. Tuan et al. try to combine data obtained from the next generation sequencing with drug susceptibility test of four commonly used antibiotics in the treatment of H. pylori.  How can authors justify using NGS instead of analyzing just genes known from previous studies that are involved in the resistance to the particular antibiotics? Did any of previously sequenced H. pylori strain was used as a control for DSTs?

The quality and clarity of the figures presented in the manuscript are questionable, and it impedes reading and fully understands the manuscript. That is why I strongly recommend improving three figures presented in the manuscript. 

However, the data showed in the manuscript has a scientific value. Strains of H. pylori that infect the population worldwide may differ a lot between each other and the strategy of combating the pathogen may be diverse among different regions of the world. That is why such studies bring benefits in developing strategy in the treatment of H. pylori.

Author Response

Response to comments raised by Reviewer 2

Comment 1. The research group isolated 55 strains from the patients, but in the material and method section, however, there is absent of information from how many patients samples were obtained.

Response:  We thank you for your comment. We added the number of patients from whom the samples were collected. Please kindly find this information in the Materials and Methods section on page 3, lines 115-116.

Comment 2. Tuan et al. try to combine data obtained from the next generation sequencing with drug susceptibility test of four commonly used antibiotics in the treatment of H. pylori.  How can authors justify using NGS instead of analyzing just genes known from previous studies that are involved in the resistance to the particular antibiotics?

Response: We thank you for your question. In fact, we used next-generation sequencing (NGS) technology to analyze genes known to confer antibiotic resistance instead of using classical approaches like Sanger sequencing-based (SS) or real time-PCR-based approaches (RT-PCR). The advantage of NGS is that it provides a more comprehensive approach, allowing us to screen all genes of interest (e.g., all genes involved in resistance mechanisms, virulence factors of H. pylori) from a given organism in their full-length within a draft complete genome. SS or RT-PCR methods can usually only provide information related to a fragment or a specific mutation of a targeted gene. In addition, it is well known that high-throughput sequences from NGS allow substantial increases in sensitivity and accuracy for detecting allelic variations compared to traditional approaches. Therefore, classical methods are less sensitive and might miss interesting information related to rare resistance mechanisms compared to NGS, as mentioned in page 2, lines 90-95, and they might limit the discovery of novel mutations. Our study showed that NGS applied at the genome level showed high capability of detecting genetic determinants given the high correlation between observed genetic determinants from NGS and the agar dilution method, which is considered as the gold standard for antibiotic susceptibility testing. 

Comment 3. Did any of previously sequenced H. pylori strain was used as a control for DSTs?

Response: We thank you for your comment. As mentioned in the Materials and Methods section page 3-4, lines 141, 158-159 and 170, the reference strain 26695 (NC_000915.1) was used as a control for DSTs of both genotype and phenotype.

Comment 4. The quality and clarity of the figures presented in the manuscript are questionable, and it impedes reading and fully understands the manuscript. That is why I strongly recommend improving three figures presented in the manuscript. 

Response: We thank you for your valuable recommendation. We already improved the three figures by changing contrast colors and increasing the resolution (300 dpi and width/height at least 1000 pixels). New figures were shown at page 6 (Figure 1), page 6 (Figure 2) and page 7 (Figure 3). We also change all fonts to Palatino Linotype to ensure consistency throughout the manuscript.

Round  2

Reviewer 1 Report

The authors have addressed each of my concerns and I am happy to recommned acceptance of this manuscript in its present form with just two minor corrections required. 

Lines 115-116 replace with This cross-sectional study included 206 consecutive outpatients (88 males and 188 females, age ± standard deviation 45.3 ± 15.3, age range 17-82 years) who had not been treated for H. pylori  infection previously.

Lines 128-129 replace with A total of 55 H. pylori clinical strains from 55 separate patients were successfully  isolated from gastric biopsy specimens.

It would also be useful if the details (gender, age range etc) were included for the patients that H. pylori were isolated from. 

Author Response

Response to comments raised by Reviewer 1

Comment 1. Lines 115-116 replace with This cross-sectional study included 206 consecutive outpatients (88 males and 188 females, age ± standard deviation 45.3 ± 15.3, age range 17-82 years) who had not been treated for H. pylori  infection previously.

Response: Thank you for your advice. We corrected this sentence in line 116, page 3.

Comment 2. Lines 128-129 replace with A total of 55 H. pylori clinical strains from 55 separate patients were successfully  isolated from gastric biopsy specimens.

Response: Thank you for your advice. We corrected this sentence in line 128, page 3.

Comment 3. It would also be useful if the details (gender, age range etc) were included for the patients that H. pylori were isolated from. 

Response: Thank you for your comment. We added this information in lines 128-129, page 3).

Reviewer 2 Report

Thank you for answering my comments and questions. I accept them all with big pleasure.

I have only one small concern about line 128 and 129: 

"A total of 55 H. pylori clinical strains from each patient were successfully

isolated from gastric biopsy specimens."  

is the word "each" necessary here or it changes the point of the sentence?

Author Response

Response to comments raised by Reviewer 2

Comment 1. I have only one small concern about line 128 and 129: “A total of 55 H. pylori clinical strains from each patient were successfully isolated from gastric biopsy specimens.” I the word “each” necessary here or it changes the point of the sentence?

Response: Thank you for your comment. As pointed out by Reviewer 1, readers might be confused about whether 55 H. pylori strains were isolated from separated patients, or it was originated from multiple strains of the same patients. We corrected this sentence according to Reviewer 1’s suggestion in lines 128-129.